# Treatment of a Paroxysmal Atrioventricular Block by Implantation of a Bipolar, Single-Chamber Cardiac Pacemaker in a Donkey

**DOI:** 10.3390/ani13172724

**Published:** 2023-08-27

**Authors:** Frederik Heun, Tobias Niebuhr, Alvaro Gutierrez Bautista, Felix Wiedmann, Nicole Verhaar, Sabine Kästner, Karsten Feige, Constanze Schmidt

**Affiliations:** 1Clinic for Horses, University of Veterinary Medicine Hannover, Foundation, Buenteweg 9, 30559 Hannover, Germany; 2Clinic for Small Animals, University of Veterinary Medicine Hannover, Foundation, Buenteweg 9, 30559 Hannover, Germany; 3Department of Internal Medicine III, University Hospital Heidelberg, Im Neuenheimer Feld 410, 69120 Heidelberg, Germany

**Keywords:** bradycardia, syncope, equine, horse, collapse, cardiology

## Abstract

**Simple Summary:**

A two-year-old donkey showed repeated signs of collapse. Detailed examinations identified an arrhythmia as the cause. This special form of arrhythmia is called paroxysmal atrioventricular block in human medicine. In this type of arrhythmia, the contraction of the heart chambers is missing in certain phases. The paroxysmal AV-block episodes were terminated by ventricular escape beats. This type of arrhythmia has not yet been described in donkeys or horses. For therapy, a cardiac pacemaker was implanted to prevent the donkey from collapsing again. Detailed information on the implantation procedure, the anesthesia protocol, and the follow-up is presented. The donkey was in good health over 17 months of follow-up and has not collapsed since.

**Abstract:**

Case Summary: A two-year-old donkey presented with recurrent syncope. Electrocardiography revealed periods without any atrioventricular conduction and without any ventricular escape rhythm with a duration of up to one minute. Finally, atrioventricular conduction resumed spontaneously with a preceding ventricular escape beat. Laboratory tests and echocardiography identified no reversible cause. The diagnosis of a paroxysmal atrioventricular block (PAVB) was made. Therefore, a single-chamber cardiac pacemaker was implanted under general anesthesia. The device was programmed in the VVI mode to prevent further syncope. The therapy was considered successful as the donkey revealed no further syncope during the follow-up period of 17 months. Clinical relevance: Clinically relevant bradycardia is rare in equids. This is the first report to our knowledge to describe a PAVB, a term commonly used in human medicine, in a donkey. Detailed information about the diagnosis and the successful therapy is included, with a special focus on the implantation and programming of the permanent pacemaker.

## 1. Introduction

First- and second-degree atrioventricular block (AVB) are common findings in equids, mostly without clinical relevance, whereas third-degree AVB is considered pathological [1]. The latter is characterized by a complete absence of conduction through the AV node and a resulting dissociation of atrial and ventricular systole [2]. In human medicine, there are further subgroups for intermediate forms and subtypes of AVB. The paroxysmal atrioventricular block (PAVB) is characterized by its spontaneous onset and return to sinus rhythm as well as long phases of physiological sinus rhythm [3]. During an episode of PAVB, there is no atrioventricular conduction and no ventricle systole, and patients suffer from recurrent syncope.

Cardiac pacemaker implantation is the treatment of choice in cases of severe, chronic bradycardia when reversible causes have been ruled out [2]. Pacemaker implantation was described in horses with sick sinus syndrome [4], in donkeys and horses with third-degree AVB [5,6,7], and in healthy horses [8]. However, no reports have been published on donkeys with PAVB. The short-term success was not uniform and was sometimes complicated [5,8]. The most frequent complications were infections at the implantation site or displacement of the lead tip inside the heart [5,8].

Cardiac pacemakers are characterized by the number and position of their intracardiac electrodes: Dual-chamber pacemakers are equipped with one electrode in the right atrium and the other one in the right ventricle. These pacemakers have the ability to synchronize the contraction of the atrium and ventricle [4]. Rate-adaptive pacemakers can increase the heart rate during exercise or stress. The detection of stress/activity can be achieved by mechanical motion sensors [4] or by local impedance measurements of the myocardium [7]. In contrast to a dual-chamber pacemaker, only one electrode is needed for single-chamber pacemakers. The latter devices can prevent syncope and are less invasive [5]. Closed-loop stimulation pacemakers combine the advantages of one electrode and the adaptation of the heart rate. The first successful implantations in donkeys were described recently [7]. The implantation procedure was previously described as occurring through the cephalic vein or the external jugular vein [5,8].

This case report describes a paroxysmal atrioventricular block and its successful treatment by implantation of a single-chamber pacemaker in a donkey.

## 2. Case Description

### 2.1. Case Presentation

A two-year-old male donkey (144 kg bodyweight) presented with a history of recurrent syncope. This symptom appeared up to 16 times per day during a half-year period of clinical surveillance. The animal was kept in a straw-padded box stall and fed hay ad libitum.

### 2.2. Investigations

The clinical examination revealed an inconstantly reoccurring bradycardia. The neurological and orthopedic examination revealed no abnormalities. Hematology and blood biochemistry including troponin I, calcium, potassium, and magnesium were within normal limits.

Telemetric long-term electrocardiography (TeleVet 100 (Engel Engineering Services GmbH, Heusenstamm, Germany); modified base–apex ECG) revealed intermittent episodes without atrioventricular conduction and long periods with normal sinus rhythm. The episodes of PAVB (Figure 1) were initiated spontaneously or by mild tachycardia. In one-third of the PAVB episodes, a ventricular escape beat was observed at the end of an episode. After single episodes of PAVB, atrial tachycardia was detected in the ECG for 20–40 s. The PAVB episodes lasting longer than 25 s correlated with dull mentation and staggering, while episodes lasting longer than 35 s were typically associated with collapse (confirmed by telemetric ECG and observation). Interestingly, no idioventricular rhythm was observed during any of the PAVB episodes. The described bradycardia occurred multiple times per day with variable duration (up to 16 events per day with a duration between 40 and 55 s). Additionally, frequent single or double second-degree AVB of Mobitz type II was visible.

The echocardiography (Vivid™ S70, GE Healthcare GmbH, Solingen, Germany), including 2D, color flow, M-mode, and tissue Doppler imaging showed only mild pulmonary valve regurgitation, so the ultrasound could not reveal a cause for the bradycardia.

The diagnosis of an idiopathic PAVB was made. Therefore, the implantation of a permanent cardiac pacemaker was indicated to prevent syncope. The pacemaker (St. Jude Medical™, Endurity™ Core, Abbott Laboratories, Inc., Saint Paul, MN, USA) and the bipolar, steroid-eluting, active fixation electrode (St. Jude Medical™, Tendril™ STS, 65 cm) were implanted under general anesthesia in dorsal recumbency after surgical preparation of the implantation site.

## 3. Treatment

### 3.1. Pre-Anesthetic Preparation

The donkey was instrumented with a telemetric surface ECG (TeleVet 100) and a temporary pacemaker for emergency maintenance of regular heart rhythm and adequate blood pressure during anesthesia. For this purpose, a bipolar electrode (TB II X 2/6 F, Osypka Medical GmbH, Rheinfelden, Germany) was inserted (Exacta^®^, Merit Medical^®^, South Jordan, UT, USA) into the right jugular vein after aseptic preparation. The tip was positioned near the apex of the right ventricle under echocardiographic control. The temporary pacemaker (Pace 101H, Osypka Medical GmbH) was programmed to the VVI mode (heart rate of 40 bpm, sensing amplitude 3 mV, pacing amplitude 6 V).

### 3.2. Anesthesia

Preoperative acepromazine (0.05 mg/kg i.m.; Tranquisol^®^, CP-Pharma Handelsgesellschaft mbH, Burgdorf, Germany), amoxicillin (10 mg/kg, i.v.; Belamox^®^, bela-pharm GmbH & Co. KG, Vechta, Germany), gentamicin (6.6 mg/kg, i.v.; Genta 100, CP-Pharma), and flunixin-meglumine (1.1 mg/kg, i.v.; Flunidol^®^ RPS, CP-Pharma) were administered. The donkey was sedated with dexmedetomidine (3.5 µg/kg, i.v.; Dexdomitor^®^, Vétoquinol GmbH, Ravensburg, Germany) and butorphanol (0.03 mg/kg, i.v.; Butorgesic, CP-Pharma). After inducing anesthesia (diazepam (0.05 mg/kg, i.v.; Ziapam^®^, TVM, Lempdes, France) and ketamine (2.5 mg/kg, i.v.; Narketan^®^, Vétoquinol GmbH)), the donkey was endotracheally intubated and positioned in dorsal recumbency. Anesthesia was maintained with sevoflurane (SevoFlo^®^, Abbott Laboratories Ltd., Maidenhead, UK) in 100% oxygen and a continuous infusion of dexmedetomidine (2.5 mcg/kg/h, i.v.). Dobutamine (initial dose 0.3 mcg/kg/min, i.v.; Dobutamine Liquid Fresenius; Fresenius Kabi Deutschland GmbH, Bad Homburg, Germany) was infused as needed to maintain blood pressure in the reference range. In the initial phase of general anesthesia, the heart rate decreased to 40 bpm and the invasively measured mean arterial blood pressure (MAP) was low (MAP 45 mmHg). Therefore, the setting of the temporary pacemaker was increased to a heart rate of 55 bpm. The MAP increased above 60 mmHg and remained between 60 and 70 mmHg over the course of anesthesia. Mechanical ventilation with a tidal volume of 10–20 mL/kg body weight and a respiratory rate of 8–12 breaths per minute was employed to maintain normocapnia (EtCO_2_ 35–45 mmHg). Total anesthesia time was 3.5 h and surgery time was 2 h.

Manually assisted recovery took place under constant ECG monitoring and O_2_ (15 L/min) insufflation. The donkey was able to stand without difficulty after 1 h of recovery time.

### 3.3. Surgical Procedure

The surgical approach to the left cephalic vein was performed in the left lateral pectoral groove at the level of the shoulder joint. Following a longitudinal skin incision of 7 cm, the vein was exposed and freed from the underlying tissue by blunt–sharp dissection. Ligatures were placed around the exposed vein, but not yet tied. After phlebotomy, the electrode was inserted into the vein and advanced towards the heart. Subsequently, the electrode tip was visualized and navigated through the right atrium and into the right ventricle by echocardiographic guidance (Figure 2). The electrode tip appeared hyperechogenic with an irregular structure, which enabled the differentiation from the temporary pacing electrode. The electrode was positioned close to the apex of the right ventricular septum and connected to a programmer device (Merlin™, St. Jude Medical™, Abbott Laboratories Inc., Saint Paul, MN, USA) that measured and displayed the electrical potentials derived from the lead tip. The electrode position had to be corrected twice due to a stimulation threshold higher than 2 V or electrode displacement in the direction of the outflow tract (Figure 2).

The position of the lead tip was confirmed by using electrical testing of the acute capture threshold, sensing amplitude and lead impedance. An active screw-in fixation ensured a secure fixation of the electrode tip. After this procedure, the stimulation threshold was 1.4 V at a pulse width of 0.4 ms and an impedance of 640 Ohm. The sensing signal was 5.6 mV. The position of the tip and the course of the electrode were confirmed by radiography (latero-lateral image of the thorax centered on the heart).

The lead sleeves were secured to the muscle fascia by three sutures (polyester multifilament, 0 USP), and the ligatures around the exposed vein were tied. Subsequently, a tissue pocket was created in the superficial pectoral muscle to facilitate the placement of the electrode. The pacemaker pocket, the muscle fascia, and the subcutis were sutured in two layers with polyglactin 910 2-0 USP (Vicryl, TM, Ethicon^®^, Johnson & Johnson Medical GmbH, Norderstedt, Germany) in a simple continuous pattern. The skin was closed with vertical mattress sutures (Nylon 1 USP, Dafilon, B.Braun VetCare, S.A., Barcelona, Spain). Finally, a stent was sewn over the incision, and the surgical site was covered with a padded bandage.

### 3.4. Programming

The programming of the pacemaker was confirmed and readjusted after the successful recovery of the donkey. Repeated measurements revealed that the capture threshold was 0.5 V at 0.4 ms duration. The sensing amplitude was 7.5 mV and the impedance was 540 Ohm. The following settings were selected based on these measured values: modus VVI, heart rate: 40 bpm, stimulation amplitude: 2.5 V with a duration of 0.4 ms, sensing amplitude: 2 mV, and stimulation configuration: bipolar.

## 4. Outcome and Follow-Up

Postoperatively, antibiotics (amoxicillin, gentamicin) and flunixin-meglumine were administered according to standard protocol for 7 days. Tinzaparin-sodium (50 anti-Xa IU/kg s.c., s.i.d., LEO Pharma A/S, Ballerup, Denmark) was commenced at 12 h following surgery for a duration of 7 days.

Postoperative care of the incision site and patient monitoring were performed according to standard protocol, and no complications occurred.

Telemetric long-term ECG for 4 days following surgery revealed that the pacemaker was working well. There were no ventricular premature beats detected. Whenever the ventricle rate decreased under 40 bpm, the pacemaker induced a ventricular contraction (Figure 3). The frequency of the AV block appeared unchanged during 40 days, but the pacemaker successfully prevented the occurrence of syncope. Seventeen months after implantation, the ECG revealed progression of the PAVB without AV conduction most of the time, but clinical examination and the owner’s report indicated that the pacemaker was functioning sufficiently well without any reported syncope. The internal storage of the pacemaker recorded a pacing percentage of 85 percent.

Echocardiography 40 days and 17 months after surgery showed that the lead was in its correct position with close contact with the endocardium (Figure 2). No structural changes of the endo- or myocardium and no pericardial effusion were visible in the echocardiographic examination. Radiographs confirmed the course and location of the electrode and the pacemaker (Figure 4). Left ventricular function (ejection fraction based on Simpson’s method of discs) remained unchanged during the follow-up period. At the time of follow-up, the battery was still charged for 10 years.

## 5. Discussion

Second-degree atrioventricular block commonly occurs in equids and results most times from a high vagal tone. However, in the case of repetitive and consecutively occurring AVB, the diagnosis of an advanced AVB is justified [1]. In the present case, an unusually long episode of AVB was observed. Single episodes without any ventricular contraction lasted up to 55 s. Commonly, a ventricular escape rhythm guarantees a minimal ventricular rhythm of 15 to 25 beats/minute [1]; this, however, was not the case in this donkey. In the case of previously described intermittent symptomatic bradycardia, an insufficient escape rhythm was the cause of syncope [5,7,9], whereas an escape rhythm was not present in the described case. Paroxysmal atrioventricular blocks are described in human medicine and are characterized by the spontaneous onset of a complete AVB without ventricular contractions [3]. The duration is variable, and restoration of AV conduction occurs spontaneously or after a single ventricular escape beat. This definition precisely describes the clinical findings in this case. Because of an absent idioventricular rhythm and the absence of echocardiographic signs of any structural pathology, a functional disorder was suspected. There were no signs of conduction disorders in the four-lead surface ECG. An advanced 12-lead ECG [10] might have been helpful to screen for conduction disorders but was not performed. High vagal tone could be ruled out as a cause of the pathological findings since the PAVB occurred both under stress and at rest. In addition, atrial frequency accelerated during PAVB, which would not be expected in the case of high vagal tone.

In cases of severe bradycardia, implanting a cardiac pacemaker is recommended after the exclusion of reversible causes [1,2]. Single-chamber cardiac pacemakers need one electrode and can improve life quality by preventing syncope [5], but they do not synchronize atrial and ventricular activity. Dual-chamber pacemakers are able to adapt the rhythm of the ventricle to the atrial rhythm and therefore guarantee a slightly better cardiac function because of the synchronization of the atrial and ventricular systole. However, the atrial electrode apparently has a higher risk of dislocation [8]. In third-degree AVB, a dual-chamber pacemaker could be beneficial to ensure atrial and ventricular synchrony. Nonetheless, the presented donkey usually had a normal AV conduction. To prevent syncope in the short phases of PAVB, a single-chamber pacemaker is sufficient. Here, the risk of complications is lower, and the battery life is longer, which delays generator replacement and reduces the risk of electrode displacement.

The transvenous implantation of the electrode is the preferred method in equids [4,5,7,8]. According to current knowledge, the electrode was implanted through the left cephalic vein of the donkey presented. The left and right sides were found to be equally suitable [8]. The chosen vein was easily accessible, and the follow-up after the procedure revealed no complications regarding wound healing and the comfort of the donkey.

The implantation procedure was described in standing sedated horses [8] and under general anesthesia [5]. In the present case, we decided to perform the procedure under general anesthesia to avoid the risk of collapsing and to guarantee the performance of the surgical procedure under aseptic conditions. Due to the high risk of bradycardia and consequently insufficient arterial blood pressure, a temporary pacemaker was used as previously described [4]. This helped to maintain heart rate and blood pressure in the event of bradycardia.

Tinzaparin was administered to reduce the risk of thrombosis at the insertion site of the venous catheter. Treatment was started 12 h after surgery to reduce the bleeding risk. Although this drug has not been administered in similar cases in donkeys so far, we decided to use it for thrombosis prophylaxis. No clinical signs of complications or thrombosis were seen.

Postoperative echocardiography is recommended to identify electrode displacement or infection of the electrode [8]. The present case showed an uncomplicated outcome during a 17-month follow-up period. Echocardiography and X-rays confirmed that the position of the electrode tip remained unchanged. Additional hematology and clinical surveillance showed no signs of inflammation. The follow-up revealed an increase in the duration of the impaired AV conduction. This progression is also described in human cases of PAVB [11].

A steroid-eluting electrode tip was used to suppress local inflammation at the implantation site of the myocardium. In this case, this may have contributed to a reduced increase in the stimulation threshold due to inflammation or fibrosis [8].

Electrophysiological measurements (lead impedance, stimulation threshold, sensing amplitude, and intracardiac electromyogram) to confirm the electrode position inside the heart were crucial for the procedure and effectively supplemented the echocardiographic and X-ray imaging. Both the character and shape of the derived potential on the electrode tip were indicative of the position inside the heart. Morphology of the R-waves and repetitive measurements of the impedance and capture thresholds indicated a stable intracardiac electrode position with optimal contact of the lead tip with the myocardium. The active fixation electrode caused damage to the endocardium and myocardium, which resulted in transient alterations in the capture threshold and resistance of the tissue. Therefore, it was necessary to perform the final programming of the cardiac pacemaker some hours after the implantation and the donkey’s recovery.

## 6. Conclusions

This case report describes a subtype of symptomatic bradycardia in a donkey and recommends the term PAVB. The characteristics of this bradycardia are strictly derived from human medicine, and we recommend using this term in veterinary medicine as well. Important criteria for diagnosis are described, and the follow-up indicates a possible progression of this disease. Successful therapy in this case was the implantation of a permanent pacemaker. A detailed description of the implantation procedure and the necessary programming is included.

## Figures and Tables

**Figure 1 animals-13-02724-f001:**
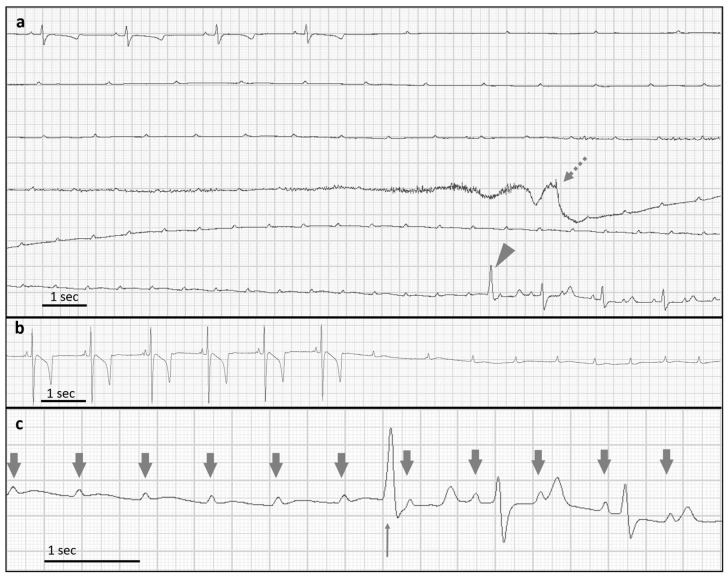
Base–apex ECG (**a**–**c**) with paroxysmal atrioventricular block (PAVB) of a donkey. (**a**) Spontaneous onset of PAVB and restoration of atrioventricular conduction after a duration of 41 s. The episode of PAVB ended with a ventricular escape beat (arrowhead). During the PAVB, the atrial frequency was accelerated. Motion artifacts indicate a syncope (dotted arrow). (**b**) Initial section of a PAVB. (**c**) Final section of a PAVB with one ventricular escape beat (small arrow). There was a 2:1 AV conduction after restoring AV conduction (large arrows indicating p waves).

**Figure 2 animals-13-02724-f002:**
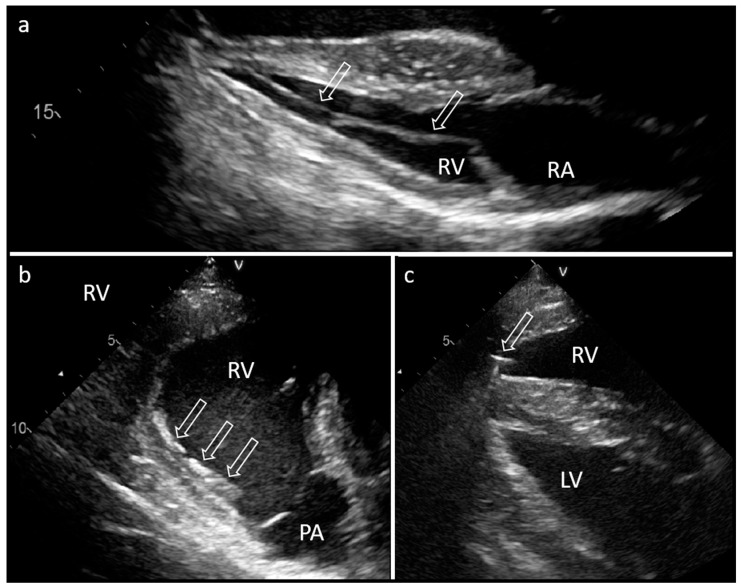
Exemplary images of echocardiography (B mode) 17 months after implantation (**a**) and during electrode placement (**b**,**c**) of a pacemaker in a donkey. (**a**) Transducer position: left body side, 4th intercostal space. The pacing electrode (arrows) is visible inside the right ventricle and is fixed at the septum near the apex. (**b**) Transducer position: right body side, 4th intercostal space. Image of the displaced electrode tip (three arrows) towards the pulmonic artery (PA) during implantation. (**c**) The electrode tip (arrow) inside the right ventricle (RV) is visible close to the fixation screw. RA, right atrium; LV, left ventricle.

**Figure 3 animals-13-02724-f003:**
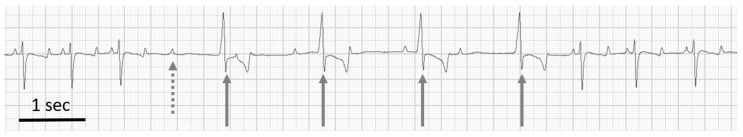
Base–apex ECG after pacemaker implantation in a donkey. The dotted arrow indicates the start of an atrioventricular block episode. The pacemaker triggered subsequent chamber depolarizations (solid arrows). Sinus rhythm reoccurred after four pacemaker-induced heartbeats.

**Figure 4 animals-13-02724-f004:**
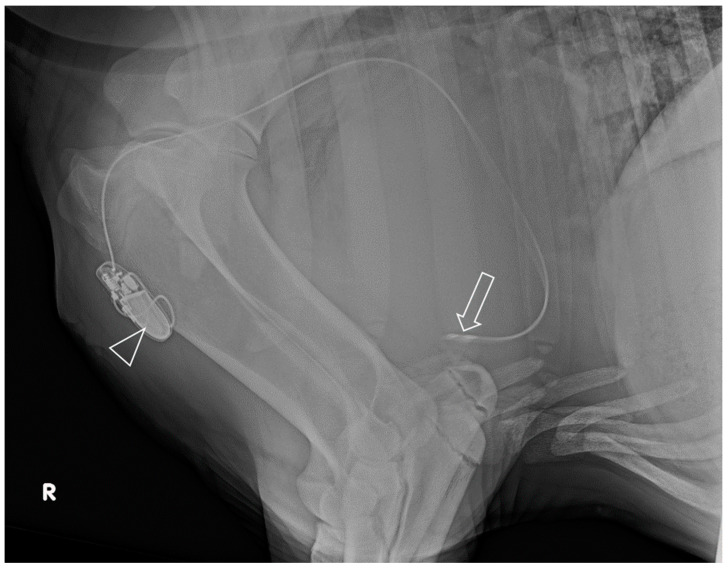
Latero-lateral radiograph of the cranio-ventral thorax of a donkey after pacemaker implantation. The lead tip (arrow) is located inside the right ventricle near the apex of the heart. The electrode course via the cephalic vein to the pacemaker (arrowhead) within the pectoral muscle is visible.

## Data Availability

All necessary data are included in the case report. For further information, please contact the author.

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
