# Peer review of "Treatment of a Paroxysmal Atrioventricular Block by Implantation of a Bipolar, Single-Chamber Cardiac Pacemaker in a Donkey"

_animals, 2023, doi:10.3390/ani13172724_

Round 1

Reviewer 1 Report

Dear Authors, very important paper which describes the possibilty to treat the third degree atrioventricular block in donkeys and horses.

Reviewer 2 Report

Congratulate with your results and description of the use of a single chamber pacemaker in a donkey.

I only have a few points that might improve your case description.

·       Did you use invasive monitoring of blood pressure during the operation?

·       Did you consider omitting the temporary rescue pacewire and instead use external pacing during the operation? It can be a little more complicated to insert pace wires at the same time as a temporary wire is in place.

·       I suppose the donkey was grown up and that the lengths of the wires were suitable for the rest of the donkey’s life, but I suggest describing the length

·       I think many donkey owners do not afford such an operation, is it expensive?

·       Estimated life of the device? I know it depends on many things, but the reader may not know, 7 -10 years?

·       I money was not a problem, it might be even better to insert a lead less pacemaker in a case like yours (1)

1.                      Bernard ML. Pacing Without Wires: Leadless Cardiac Pacing. Ochsner J. 2016;16(3):238-42.

Best regards

Reviewer 3 Report

The topic of utilizing a pacemaker for managing paroxysmal atrioventricular block (PAVB) in a donkey is intriguing and expands our understanding of veterinary cardiology. This case study is expected to make a valuable contribution to the existing literature in the field.

I believe that your article, with its detailed investigation and description of the treatment process, provides important insights for veterinary practitioners and researchers. The case of a donkey with PAVB is of particular interest due to the few cases of implementing a pacemaker in equids, and its potential impact on improving animal care.

Could you please provide the following details to enhance the article's completeness:

2.2. Investigations

line 85-86. The blood biochemistry included calcium and magnesium? Could you mine specify if these electrolytes were included?

Reviewer 4 Report

This paper is a nice overview of the diagnosis and treatment of paroxysmal AVB in a donkey with a permanent cardiac pacemaker. In general, the paper is well written, and the authors thoroughly describe the case presentation and surgical technique. Details of the specific pacing device and settings are included – which are lacking in some other reports of permanent pacing devices in equids.

Although I enjoyed reading this article, it lacks originality. The specific term ‘paroxysmal atrioventricular block’ was not used previously, however, intermittent periods of sudden AVB without a reliable ventricular escape rhythm have been reported previously in donkeys (including one published in this journal in 2021). Additionally, the specific characteristics of this pacing device (bipolar, steroid-eluding, single chamber, active fixation) have been described previously. The follow-up period appears insufficient considering the previous reports of endocarditis occurring 2 and 3 years after implantation in a previous study of permanent pacing devices in donkeys (De Lange et al. 2021.) 

There is some awkward English language use, including inappropriate use of plurals, throughout the paper including the simple summary and abstract. Therefore review by an English language specialist is recommended.

Round 2

Reviewer 4 Report

I thank the authors for the time and detailed responses to my previous comments, however, my concerns over the lack of novelty of this diagnosis remain.

The conclusions are much improved from the previous version and contain only minor errors in English – ‘these’ (line 302) is plural and should be singular to match the singular ‘bradycardia’.

Line 40 - Although the term ‘paroxysmal atrioventricular block’ was not used, periods of sudden complete heart block without a ventricular escape rhythm have been reported previously in donkeys. The authors’ descriptions of the previously published reports do not provide any additional insight.

Line 59-60 – Again, although the term PAVB wasn’t used, this condition has been described in donkeys.

Line 91, 105 – The issue remains that a ventricular complex which occurs after a long pause is not ‘premature’. Although the authors have cited a reference which also incorrectly uses the term premature, the terminology is not correct.
For the correct use of the term escape beat/complex, please see references (among others) below:

Ferasin, L., et al. "Syncope associated with paroxysmal atrioventricular block and ventricular standstill in a cat." Journal of small animal practice 43.3 (2002): 124-128. (Figure 3)

Komatsu, Sayaka, et al. "A proposal of clinical ECG index “vagal score” for determining the mechanism of paroxysmal atrioventricular block." Journal of Arrhythmia 33.3 (2017): 208-213. (Figure 3)

Lee, Sinjin, Hein JJ Wellens, and Mark E. Josephson. "Paroxysmal atrioventricular block." Heart Rhythm 6.8 (2009): 1229-1234. (Figure 3)

Silvetti, M. S., et al. "Paroxysmal atrioventricular block in young patients." Pediatric cardiology 25 (2004): 506-512. (Figure 1)

deSouza, Ian S., and Monisha Dilip. "Fortuitous Identification of Fluctuating AV Block: A Case Report." The Journal of Emergency Medicine 57.1 (2019): e9-e12. (Figure 2)

Den Dulk, Karel, et al. "Myocardial bridging as a cause of paroxysmal atrioventricular block." Journal of the American College of Cardiology 1.3 (1983): 965-969. (Figure 2)

Grutter, Giorgia, et al. "Paroxysmal atrioventricular block after heart transplantation in children: an early sign of rejection?." Pediatric Transplantation 20.8 (2016): 1164-1167. (Not in figure but described in text)

Although the authors state the manuscript has been reviewed by an ‘English native speaker’, no such changes have been made. Therefore, review of the manuscript by an English language specialist more familiar with the specific subject matter is recommended. This is certainly not an exhaustive list, but some examples include: inappropriate use of ‘syncopes’ (lines 32, 38, 39, 55, 70) and ‘atrioventricular blocks’ (lines 47, 48, 59) – neither of which should be plural. “Modus” (line 37) is not an English word; I think the authors mean “mode”. The phrase “the contraction of the heart chambers is missing in certain phases” (lines 26-27) needs revising. Line 51 should not begin with ‘the’.
